

# New data on tail lengths and variation along the caudal series in the non-avialan dinosaurs

David W.E. Hone[1], W. Scott Persons[2] and Steven C. Le Comber[1,†]

[1] School of Biological and Chemical Sciences, Queen Mary University of London, London, United Kingdom
[2] Mace Brown Museum of Natural History, College of Charleston, Charleston, SC, United States of America
[†] Deceased.

## ABSTRACT

The tails of non-avialan dinosaurs varied considerably in terms of overall length, total number of vertebrae, and gross form and function. A new dataset confirms that there is little or no consistent relationship between tail length and snout-sacrum length. Consequently, attempts to estimate one from the other are likely to be very error-prone. Patterns of changes in centra lengths across the caudal series vary among non-avian dinosaurs. However, some overarching patterns do emerge. A number of taxa show (anterior to posterior) a series of short centra, followed by a series of longer centra, with the remainder of the tail consisting of a long series of centra tapering in length. This pattern is consistent with functional constraints, and the anterior series of longer centra are coincident with the major attachments of femoral musculature. This pattern is not present in many basal taxa and may have evolved independently in different dinosaurian groups, further suggesting functional importance.

## INTRODUCTION

The caudal vertebral series of the non-avialan dinosaurs (hereafter simply 'dinosaurs') were variable in form (Fig. 1) and served many roles. Dinosaur tails had a biomechanical function in locomotion (e.g., *Hutchinson, Ng-Thow-Hing & Anderson, 2007*; *Persons & Currie, 2011a*) and balance (e.g., *Hutchinson & Gatesy, 2001*; *Libby et al., 2012*), and some were specialized for behavioural roles including inter- and intraspecific combat (e.g., *Mallison, 2011*; *Arbour, 2009*) and signaling (e.g., *Persons, Currie & Norell, 2014*). Despite this importance, the osteological caudal anatomy of dinosaurs has received far less attention than most other major anatomical regions.

To date, few dinosaur tails have been identified that are truly complete (i.e., represented by every single vertebra in the series). Those that are complete show considerable variation both inter- and intraspecifically (*Hone, 2012*). This limits potential comparisons between taxa and the confidence of attempts to reconstruct tail forms for taxa that lack substantial caudal material, especially estimations of aspects like total length (*Hone, 2012*).

Studies of dinosaur tails have often focused on tails as flexible structures (*Pittman et al., 2013*) to support large muscle groups, in particular the caudofemoralis, which serves as

Corresponding author
David W.E. Hone,
dwe_hone@yahoo.com

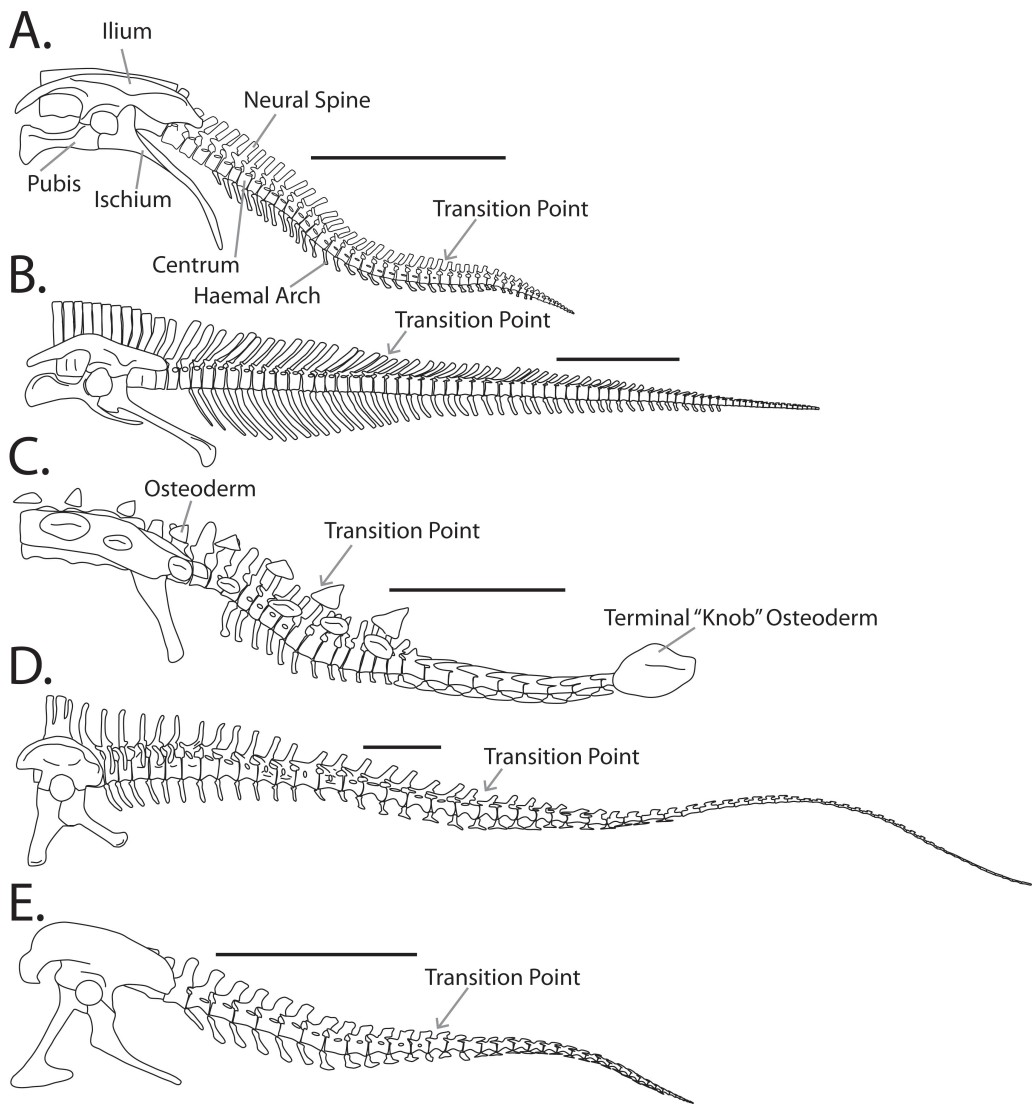

**Figure 1** **Example tail forms across the Dinosauria.** (A) Anklosaurid (*Euoplocephalus tutus*), (B) ceratopsid (*Centrosaurus nasicornis*), (C) hadrosaurid (*Corythosaurus intermedius*), (D) diplodocid (*Diplodocus carnegii*), (E) tyrannosaurid (*Gorgosaurus libratus*). Scale bar is 1 meter.

a major driver in locomotion (*Allen, Paxton & Hutchinson, 2009*; *Persons & Currie, 2011a*; *Persons & Currie, 2012*). Previous work has argued that the caudofemoralis has substantially influenced the form of anterior caudal osteology, and adaptations suggested to be linked to the caudofemoralis include haemal spine depth, prominent chevron and vertebral sulci (e.g., *Persons & Currie, 2011a*; *Persons & Currie, 2011b*; *Persons & Currie, 2014*; *Cau & Serventi, 2017*), and, most frequently, the 'transition point' of the lateral processes (*Russell, 1972*; *Gauthier, 1986*; *Gatesy, 1990*; *Gatesy, 1995*; *Persons & Currie, 2011b*). The transition point is the region of the tail where the lateral processes end, and, by extension, where the caudofemoralis is inferred to have terminated. The association of a functionally-distinct major muscle set with one region of the tail suggests that dinosaur tails were modular,

with different tail regions functioning in different ways. Modularity is obviously true of dinosaurs that bear highly-derived caudal features, such as pygostyles or tail-clubs. Such derived and obvious specialisations notwithstanding, when compared to vertebrae from elsewhere in the axial column, caudal vertebrae are generally simple in overall form. Dinosaur tail modularity may be true more generally and be associated with less apparent morphological variations.

Based on these considerations, this paper focuses on variation in anteroposterior centrum length. Little previous work has been done examining patterns of centrum lengths, although centrum length has obvious influence on tail structure and, by extension, tail function. Recently, *Nuñez Demarco et al. (2018)* demonstrated a general pattern of either decreasing lengths of vertebrae along the tail, or an increase in length to the midpoint, followed by a decrease for a number of extant reptiles. A reduction in the length of caudal vertebrae along the distal tail of the extinct *Mesosaurus* was also demonstrated, though some individuals showed stability in lengths distally (*Nuñez Demarco et al., 2018*).

Assuming otherwise equivalent form, a series of long vertebrae would, as a unit, provide relative stiffness and stability to a tail (or at least parts of it), while a series of shorter vertebrae would provide a zone of greater relative flexibility (e.g., see *Persons, Currie & Norrell, 2014*). Having short vertebrae means having more flexing points per unit of length. All other factors remaining equal, a three-meter-long section of tail with thirty vertebrae is more flexible than a three-meter-long section of tail with only twenty vertebrae.

It has been argued that the proportions of dorsal vertebrae are good correlates of absolute body size in tetrapod (see *Currie, 1978* for a review), but no similar argument has been made for caudal vertebrae. The total number of vertebrae is correlated with body size in many basal vertebrates (*Head & Polly, 2007*), but less so in taxa where there is vertebral regionalization and functional constraint (as in birds and mammals—*Wake, 1979*). The tails however, may vary considerably, even in mammals (both in terms of caudal count and total length (e.g., *Garland Jr, 1985*; *Cavallini, 1995*; *Alroy, 2019*) suggesting it is relatively free of such constraints.

In dinosaurs, changes in vertebral length down the caudal series have typically been considered and described as simple sequential reductions (e.g., see *Gilmore, 1936* on *Apatosaurus* and *Sereno, 1987* on *Psittacosaurus*). However, the proximal caudal vertebrae of *Apatosaurus* actually show sections of increase (in terms of both proportional and absolute centrum lengths—see below), and in the tail of *Psittacosaurus*, although the vertebrae never increase in absolute length, there are sections of length stability and decreases are not always regular (*Sereno, 1987*).

In this context, we hypothesise that there is high variation in tail length across the Dinosauria, which make total body-length difficult to predict (following *Hone, 2012*). We would also expect the tail structure of bipedal taxa differs from that of quadrupeds, as the tail is more important for balance in bipeds. We predict that centrum length does not follow a simple pattern of decrease in length in successive vertebrae. Specifically, the presence of the lateral processes is known to be associated with a major change in tail function (the presence of the caudofemoralis musculature) and we, therefore, predict that the posterior loss of the lateral processes correlates with a change in centrum lengths. It is

expected that testing these hypotheses will reveal variation that is important to the function and evolution of tails in different subclades and locomotor regimes within Dinosauria.

## MATERIALS & METHODS

We expanded on the dataset of *Hone (2012)*, with additional measurement data collected directly from specimens, from photographs, and from the literature. Previously overlooked and new material was identified by ourselves and also suggested by various sources (see acknowledgements).

A complete tail was defined as one with every vertebra present down to the last caudal. The last caudal can typically be identified by a rounded posterior face (this trait remains useful even in procoelous vertebrae, because the extent of the rounding exceeds that observed elsewhere in the series) and a lack of postzygopophyses and/or neural spine (*Hone, 2012*). Additional tails were regarded as complete where, although one or more elements were not preserved, the absent material could still be accurately recorded because there was either an impression of the missing material in the matrix or the missing material was bounded by other elements. Nearly complete tails were also included where it was felt that the missing material could be accurately reconstructed from other specimens of the same genus or species. For example, a total tail length was calculated and included where two or more individual specimens were complete enough to suggest the animals had very similar body sizes, and where the tails of both included a series of overlapping elements (e.g., an anterior tail portion and a posterior tail portion, with both possessing the last chevron or last lateral process pair). Total tail length was taken as the sum total of all individual caudal centra length measurements or as the single measurement of the caudal series when centra were preserved closely appressed together. In many cases, the live animals may have possessed intervertebral discs that would have increased tail length, but these cannot be easily estimated and so were simply excluded (though see also *Rothschild et al., 2020* who argue for their absence).

Total femoral length was taken as a simple proxy for mass/body size (following *Hone, 2012*) for each specimen. Although other proxies (e.g., femur circumference—*Campione & Evans, 2012*) are stronger correlates of mass, femur length is appropriate for such datasets, and length was the only consistently reliable measurement for samples from the literature and from specimens that are taphonomically distorted. To examine the relationship between tail length and body size, we compared snout to sacrum length and tail length using a simple linear least-squares regression. Snout-sacrum length was taken as the combined length of the skull and every cervical, dorsal and sacral centrum (see *Hone, 2012* for further details). Since in some cases measurements came from different individuals, we scaled both against the femur of the specimen from which it was measured. This was also carried out for various subsets of the data to test the hypothesis that locomotion patterns of different groups affect tail length. We looked at four broad divisions of dinosaurs based on a general understanding of their locomotion: obligate bipeds (theropods, non-sauropodan sauropodomorphs), obligate quadrupeds (sauropods, thyreophorans), bipeds and facultative bipeds (iguanodontids, hadrosaurs, psittacosaurids) together, and quadrupeds and facultative bipeds together.
Incomplete tails were not included in the analyses (with the exception of those reconstructed as described above), as it was considered impossible to ascertain the missing material based on the variations in caudal counts (see below). Even tails that appear to be tapering consistently to a tip may have some considerable length still missing as seen with *Diplodocus*, for example. However, in an attempt to maximise the limited available data, we also sourced tails that were incomplete, but considered likely to be close to complete. Such tails can at least be used to demonstrate minimum tail lengths, as an incomplete tail that is as long as or longer than a complete tail still demonstrates a genuine difference (data are provided in the Appendix).

The definition of what constitutes a 'nearly' complete tail is necessarily subjective given the limitations of the available information. The intention was to include only those judged to have very few caudals missing and only a very short amount of the tail missing in terms of length. In order to estimate this, we took into account the degree of tapering of the tail, the length of material preserved and the tail lengths of close relatives. For example, not included is the diplodocid sauropod *Barosaurus* AMNH 6341 (*McIntosh, 2005*), which has 29 preserved caudals that total over 6 m in length (against a femur of just 1.4 m in total length). However, the last preserved caudal in AMNH 6341 is 171 mm long and, while some sauropods have as few as 35 caudals (*Borsuk-Bianka, 1977*), a large amount of tail is considered likely missing, given the size of the caudals present and the considerably higher number of caudals in other diplodocids (e.g., *Gilmore, 1936*). Note that the holotype of the small hadrosaur *Tethyshadros* (*Dalla Vecchia, 2009*) was incorrectly considered complete in *Hone (2012)* and so is not in these datasets (though it is included in the section on individual centra lengths).

The patterns of individual caudal centrum lengths that make up dinosaurian tails were also analyzed. Here data from the above specimens was supplemented with additional, but incomplete, tails as the analysis looked at changes in individual centra as part of a series, rather than the tail as a whole unit. Note that even tails that can be diagnosed as complete are not always included in the analysis, since either information on individual vertebrae lengths was not available in the literature or the divisions between the vertebrae could not be reliably measured (e.g., the holotype of *Jinfengopteryx*—CAGS IG 040801).

The data is unevenly distributed with a bias towards small theropods (likely a taphonomic bias from Lagerstaetten deposits) and likely further biased due to high interest in the maniraptoran-avian transition, and with a bias against sauropodomorphs (large animals with numerous tail vertebrae that were seldom buried and preserved as complete specimens). Thus, for the data on snout-sacrum vs complete tail length, there were 16 ornithischians (of which seven were ceratopsians), 12 theropods (of which seven were deinonychosaurs), but only three sauropodomorphs.

To test the hypothesis that caudal centrum lengths do not follow a simple decrease in size along the series, we used segmented regression (also known as piecewise or broken stick regression) to identify transitions in centrum length (break points). These length transitions were then compared with the boundary between the muscular and less-muscular parts of the tail (the transition point). Essentially, this approach, where appropriate, fits a series of linear regressions to specific subsets of the data. In some cases, however, a simple linear

regression will provide a better fit to the data. Because of some debate over the best way to assess goodness of fit for segmented regressions (e.g., *Hall, Osborn & Sakkas, 2013*) the analysis was started with a Davies test (*Davies, 2002*) for a non-constant regression parameter in the linear predictor (vertebra number) on centrum size using the R package *segmented* version 0.4–0.0 (*Muggego, 2003*; *Muggego, 2008*) implemented in R version 3.0.3 (*R Core Team, 2014*). This allows for an independent test of whether a segmented regression is justified, meaning that we were able to use a segmented regression where the data support it and a linear regression where there is no evidence to support a more complex fit.

It is possible that the inevitable variations and slight inconsistencies of collecting data from specimens vs the literature or photographs may affect marginal results. Therefore, we also took one specimen (a hadrosaur—TMP 1998.058.001) as a test case for variation in measurements between first hand observations and photographs. Each caudal centrum length was measured physically 10 times, and the same specimen was then photographed and another 10 replicate measurements obtained from these photographs. Segmented regressions were then fitted in the same way as the rest of the study, to both independently estimate the break points and in particular their congruence to each other, and to estimate the transition point (vertebra 12).

Where the Davies test suggests that a segmented regression is appropriate, the next question is how many segments to fit. Candidate models with 1-4 breaks were fitted using *segmented* and the best model was selected on the basis of the Akaike Information Criterion (*Akaike, 1974*). In essence, this approach selects the model which best balances explanatory power with simplicity. An upper limit of four breaks was chosen, since some specimens had 20 or fewer vertebrae, limiting the number of breaks that could be plausibly fitted.

Models used the default parameters in *segmented*, with quantiles as the starting points for the iterative breakpoint analysis, but with 50 bootstrap samples and a maximum of 10 iterations. Candidate models that could not be fitted in *segmented* (usually because of gaps in the data) were discarded.

In one case (*Dyoplosaurus*) the Davies test indicated that a simple linear model was appropriate but visual inspection of the fit indicated that there are in fact two separate groups of vertebrae. The breakpoint models fitted by *segmented* assume a continuous relationship but in this case it was clear that a discontinuous model would be best. Therefore, one was fitted by classifying the vertebrae into two groups and then including this as a factor in a linear model plus the interaction term with vertebra number.

## RESULTS

### Tail length vs snout-sacrum length

Our hypothesis that there is high variation in tail lengths across the dinosaurs, and that tail lengths do not correlate well with body size (i.e., snout-sacrum length) was supported with very wide confidence limits for predictions of one based on the other (Fig. 2). However, the prediction that there would be similarities between bipedal or quadrupedal taxa was not met. Even when considering facultatively bipedal or quadrupedal taxa with obligate biped and quadrupeds, no clear relationship between body size and tail length was recovered.
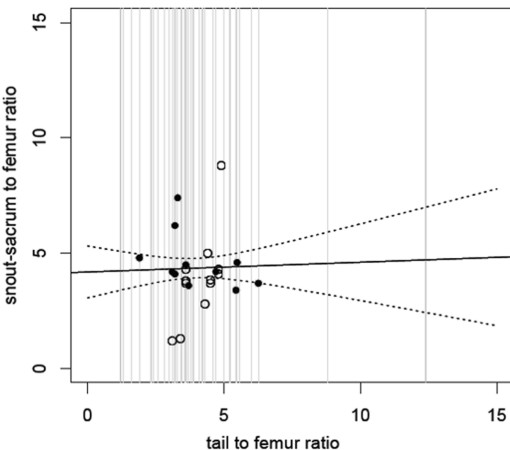

**Figure 2 Relative size of snout to sacrum against relative tail length.** Snout to sacrum vs tail length, with both measurements scaled to femur size (see Methods). Comparisons drawn from the same individual are shown as black circles; those from different individuals as open circles. The solid line shows the fitted (non-significant) regression; dashed lines show 95% confidence intervals for this regression. The grey lines show the range and distribution of tail to femur ratio for all of the species in our analysis and demonstrate that while most cluster in the centre, there are extremes (from 1.3 to 12.4) and here the confidence intervals would be even wider. See also Table 1.

The model suggests that tail length is an extremely poor predictor of snout to sacrum length across dinosaurs (Fig. 2) (linear regression of snout/sacrum to femur ratio on tail to femur ratio: $F_{1,21} = 0.11$, $p = 0.74$). For example, *Scutellosaurus* has a tail to femur ratio of 8.8, and with a femur length of 82 mm, this corresponds to an estimated snout-sacrum size between 263.0 and 485.1 mm, though the real value is much closer to the upper bound of 405 mm. At the opposite end of the scale, the tail to femur ratio of 1.2 in *Epidexipteryx* would correspond to an estimated snout-sacrum range of 173.4 to 259.1 mm although the actual value is outside of even this broad range at 158 mm.

The same poor prediction of snout-sacrum length by tail length is retained (linear regression: $F_{1,16} = 0.01$, $p = 0.91$) if we attempt to minimise problems due to intraspecific (ontogenetic) scaling or phylogenetic bias (where there are multiple specimens from some species that are therefore over-represented) and restrict the analysis to the largest individual of each species. This pattern is also true of the various subdivisions based on locomotor style: bipeds alone (linear regression of snout-sacrum to femur vs tail to femur)—$F(1,8) = 0.01$, $p = 0.92$; quadrupeds alone—$F(1,5) = 0.2938$, $p = 0.61$; bipeds plus facultative bipeds—$F(1,14) = 0.331$, $p = 0.57$; quadrupeds plus facultative bipeds—$F(1,11) = 0.009$, $p = 0.92$. In short, tail size is not clearly related to body size in non-avian dinosaurs, even allowing for broad distinctions in locomotor style.

## Patterns of caudal vertebrae length

Across the Dinosauria, most tail sequences passed the Davies test and so could be reconstructed with one or more breaks to the series of individual centrum lengths. Dinosaur tails do not show simple patterns of change in centrum lengths but typically exhibit multiple distinct regions. There is at least some consistency in the results within

clades, with several specimens of single genera showing similar patterns to one another, though others show high levels of variation within one species. The details of these results for various species and clades are considered in more detail below.

## Assessment of consistency of centrum measurements

Where multiple measurements were taken from one hadrosaur tail as a test of consistency of measurements, the congruence was good between break points produced from direct measurements of the specimen and a photograph. In both cases, the model fitted four break points (specimen (mean ± se): 12.0 ± 1.59, 45.8 ± 3.48, 61.5 ± 3.90, 74.7 ± 0.71; photographs 11.9 ± 1.99, 48.5 ± 1.29, 57.3 ± 1.07, 59.4 ± 1.04). In particular, both methods fitted a break point very close to the actual transition point, and even at the distal end, where there was slight disagreement, all four of the break points fitted from photographs were encompassed with the standard errors of break points derived from the specimen itself (Fig. 3). This suggests that the measurements taken from photographs for various specimens will yield accurate data.

## Break points

For 18 out of 25 specimens, the distance between a break point and the transition point was lower than would be expected by chance (exact binomial test, $p = 0.043$) (Fig. 4). This suggests that there is a relationship between the end of the lateral processes (and by extension the termination of the caudofemoralis muscles) and a change in length of the centra in the tail.

The results of the break point analyses are shown in Figs. 5–12. These are the non-sauropodan sauropodomorphs (Fig. 5), sauropods (Fig. 6), earlier branching theropods (Fig. 7), mairaptorans (Fig. 8), dromaeosaurs (Fig. 9), *Coelophysis* and *Archaeopteryx* (Fig. 10), thyreophorans (Fig. 11), iguanodontians and hadrosaurs (Fig. 12) and ceratopsians (Fig. 13).

Some genera are extremely conservative, such as *Archaeopteryx*, with others varying considerably, such as *Coelophysis* (both Fig. 10). Most taxa show an early short series of tail vertebrae that decrease in length sequentially, then a short series that increase in length (typically including the longest centra in the tail), followed by a long series of progressive decrease. These include at least one specimen each of *Lufengosaurus* (Fig. 5), *Apatosaurus*, *Camarasaurus* (both Fig. 6), *Gorgosaurus, Tyrannosaurus* (both Fig. 7), *Ornithomimus*, *Nomingia* (both Fig. 8), *Microraptor*, and *Velociraptor* (both Fig. 9), *Kentrosaurus* (Fig. 11), and *Leptoceratops* and *Centrosaurus* (both Fig. 13). Frequently, the first and second caudal vertebrae may vary from this pattern. While not universal, this pattern is widespread in the Dinosauria. Other specimens are close to this pattern (e.g., *Dilophosaurus* (Fig. 7), *Ingenia* (Fig. 8), *Ouranosaurus* and *Lambeosaurus* (both Fig. 12)) with for example, one extra break point in the series. It is possible that this pattern is even more prevalent but is, in some instances, hidden from the tests used here by some variation or lack of data. For example, the proximal caudals of *Diplodocus* and *Majungasaurus* are not recorded, and although those preserved seem to conform to the pattern, it cannot be confirmed. In the case of *Dyoplosaurus* the discontinuous model was the best supported (AIC simple linear
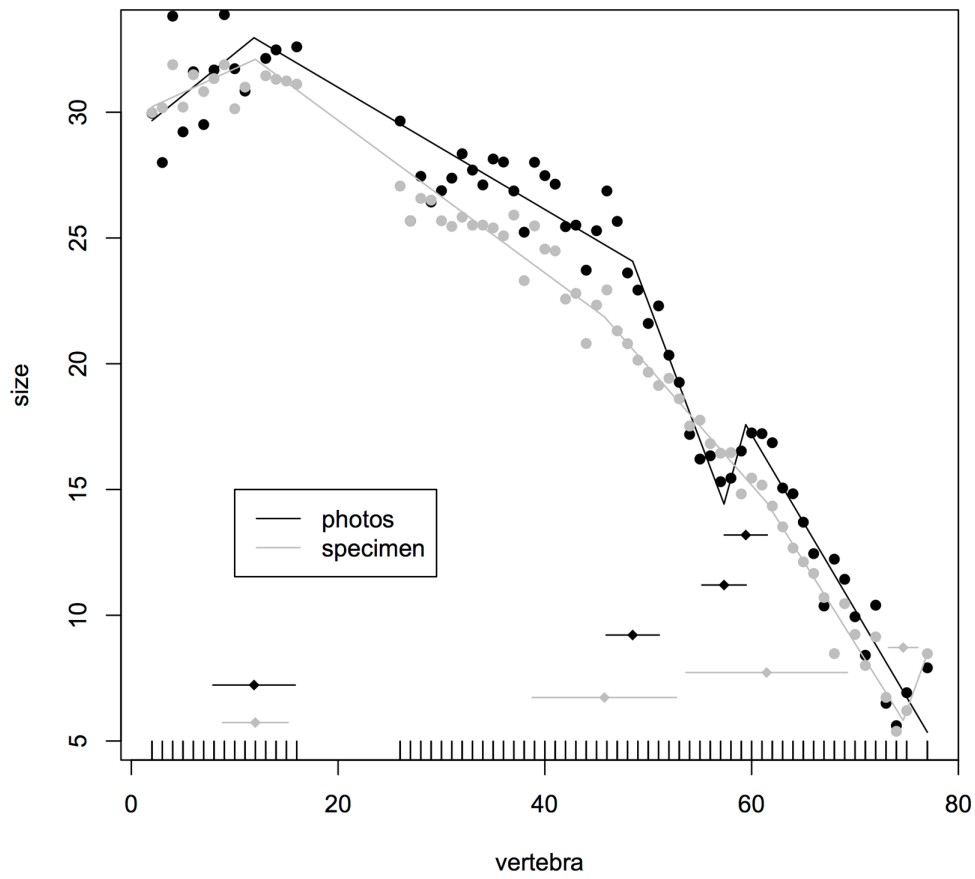

**Figure 3 Segmented regressions for the indeterminate hadrosaur specimen TMP1998.058.001.** Segments derived from photographs (red) and from the specimen itself (blue). Both methods fitted a break point very close to the actual transition point, and with the exception of a break point right at the distal end, break points derived from one method overlapped with break points derived from the other. The predicted break points and their error bars are indicated at the bottom of each graph where these have been calculated, and the transition point (where known) is indicated by an arrow.

regression. = 121.7, AIC discontinuous model = 96.0) and so this was retained (Fig. 11) with a proximal section of short vertebrae with decreasing centrum lengths and a distal segment of much longer, and near stable, centrum lengths.

## DISCUSSION

### Overall lengths of dinosaurian tails

Dinosaur tail lengths vary widely overall and the correlations between tail length and snout-sacrum lengths for complete tails is poor. The data from the 'near complete' tails is of course limited in its use given these issues, although some specimens do suggest that there may be some consistency within groups. For most dinosaur groups, attempting to estimate the total length of a tail from anything other than a near complete series is subject to a wide range of error and uncertainty, as seen even within some clades represented by few specimens (e.g., Scansoripterygidae). Dinosaur tails were also likely evolutionally

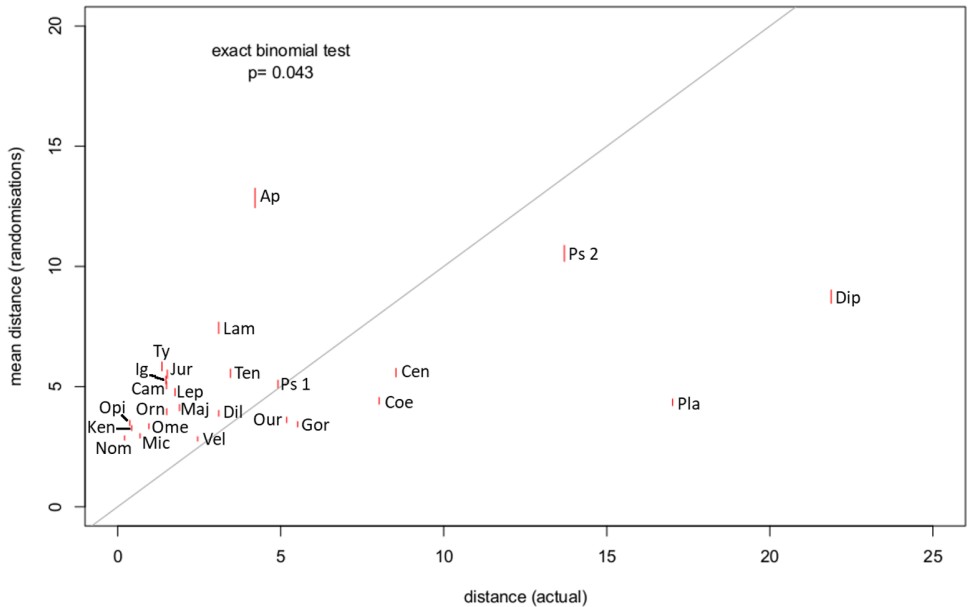

**Figure 4  Mean distance between break point and transition point in 1,000 randomisations, plotted against the actual distance.** The grey line shows the 1:1 line; red lines show standard errors. For all but seven of the specimens, the actual distance was lower than the randomised distance (Exact binomial test, $p = 0.04$). Abbreviations for listed specimens are as follows: Ap *Apatosaurus*, Cam *Camarasaurus*, Cen *Centrosaurus*, Coe *Coelophysis*, Dil *Dilophosaurus*, Dip *Diplodocus*, Gor *Gorgosaurus*, Ig *Iguanodon*, Jur *Juraventor*, Ken *Kentrosaurus*, Lam *Lambeosaurus*, Lep *Leptoceratops*, Maj *Majungasaurus*, Mic *Microraptor*, Nom *Nomingia*, Ome *Omeisaurus*, Opi *Opistocoelocaudia*, Orn *Ornithomimus*, Our *Ouranosaurus*, Pla *Plateosaurus*, Ps *Psittacosaurus* (1 = AMNH 6253, 2 = AMNH 6254), Ten *Tenontosaurus*, Ty *Tyrannosaurus*, Vel *Velociraptor*.

plastic, given the highly derived forms seen in clades, such as ankylosaurs and diplodocids. Similarly, multiple lineages show adaptation of the tail to specific functions (or specific combinations of functions) such as defense and socio-sexual signaling, adding further to interspecific, and perhaps also intraspecific, variation.

## Caudal length patterns in dinosaur tails

Considerable variation is seen not just in the overall and proportional sizes of dinosaur tails, but also in the lengths of the individual caudal centra that comprise them. Although the distalmost caudals of a series are generally smaller than more proximal ones, over a short section of consecutive elements, there may be patterns of increasing length, stability, or decreasing length (and all three may occur in one individual e.g., *Apatosaurus* CM 563—*Gilmore, 1936*). The datasets here are somewhat limited, but cover a wide range of dinosaurian biology—large and small, herbivores and carnivores, bipeds and quadrupeds, long and short tails, and taxa from multiple environments. Thus, considerable variation may be expected, but there are nonetheless some clear patterns. Most notably, many dinosaurs show repeated series of, on average, increasing and decreasing centra lengths along the caudal series as demonstrated by the positions of break points and the associated regressions.

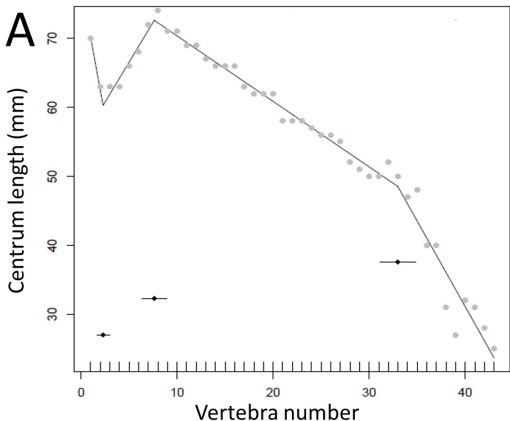

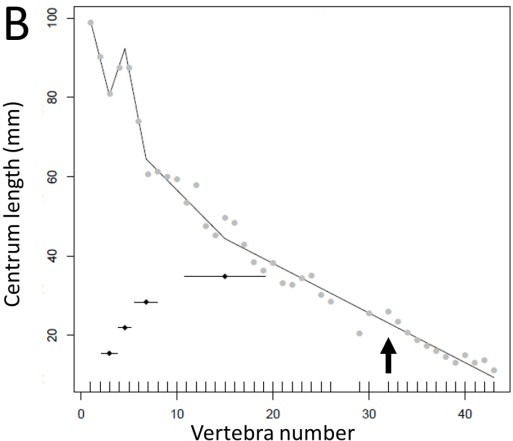

**Figure 5** Regressions for centrum lengths within the tails for non-sauropodan members of the Sauropodomorpha, (A) *Lufengosaurus,* (B) *Plateosaurus.* While different patterns are recovered, both show long sections of the tail with a tapering pattern of centra lengths. The predicted break points and their error bars are indicated at the bottom of each graph where these have been calculated, and the transition point (where known) is indicated by an arrow.

However, other taxa deviate considerably from the short-long-shortening pattern described above (e.g., *Plateosaurus* (Fig. 5), *Juravenator* (Fig. 7)); indeed the caudals of *Coelophysis* tend to increase in length for much of the series (Fig. 10). Various constraints may confound the basic pattern and affect the overall distribution. For example, the majority of the vertebrae in the dromaeosaurs *Velociraptor*, *Deinonychus* and *Microraptor* are bound by a complex series of extended zygopophyses and chevrons that stiffen the tail and perhaps free the vertebrae from normal functional constraints. This may explain some of the variation seen between specimens of a species or genera (Fig. 10). However, the patterns of centrum lengths seen in the tails of all three specimens of *Archaeopteryx* and the putative glider *Microraptor* are strikingly similar and, although only a very limited set of data, show a level of consistency often not seen in other groups (Figs. 9 and 10). This suggests that the similarity in form is connected to the shared tail function of control in flight. It has been noted by *Gatesy & Dial (1996)* that *Archaeopteryx* would benefit in

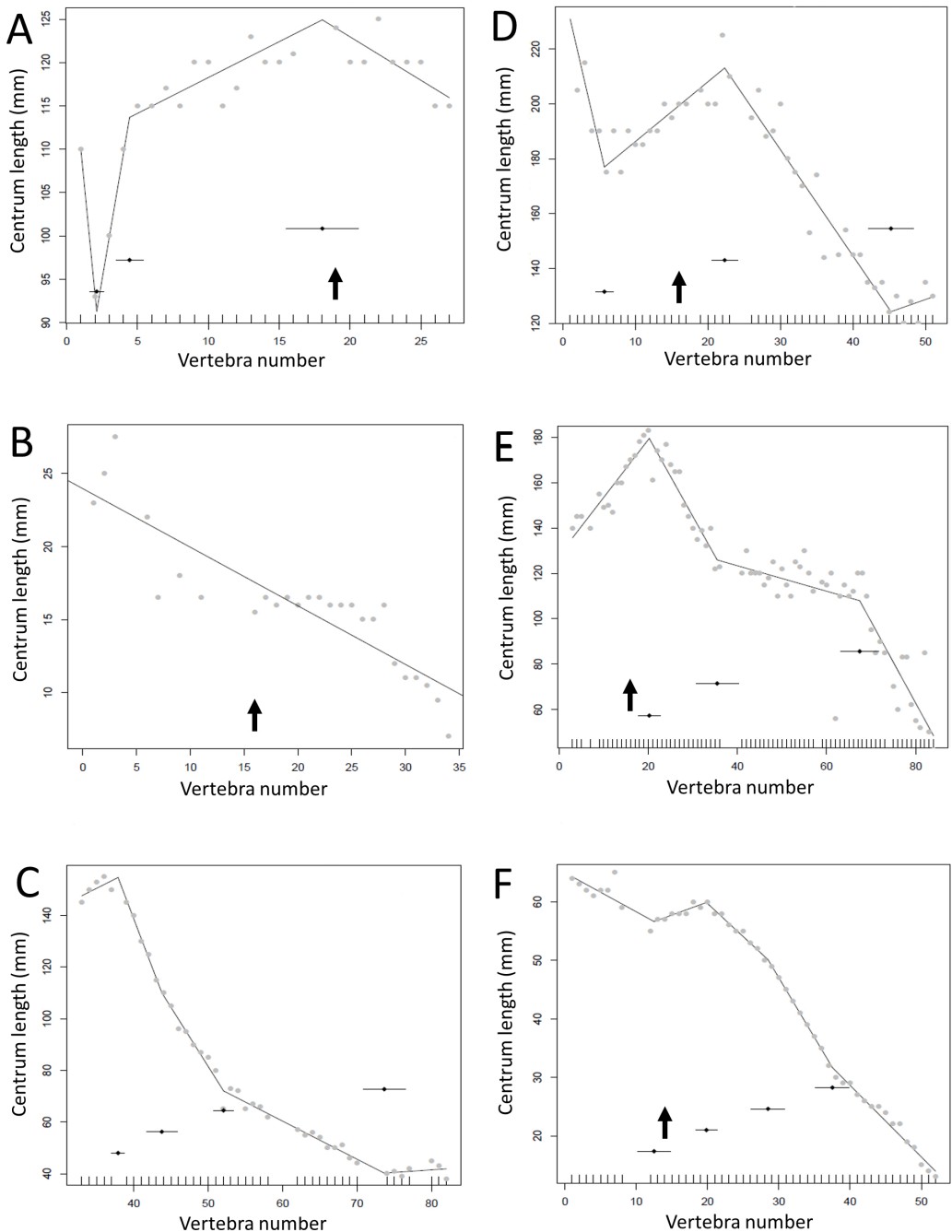

**Figure 6** **Regressions for centrum lengths within the tails of members of Sauropoda.** The predicted break points and their error bars are indicated at the bottom of each graph where these have been calculated, and the transition point (where known) is indicated by an arrow (the same point is inferred in *Apatosaurus* E based on D). (A) *Omeisaurus*, (B) *Opsitocoelocaudia*, (C) *Diplodocus*, (D) *Apatosaurus* CM3018, (E) *Apatosaurus* CM3378, (F) *Camarasaurus*.

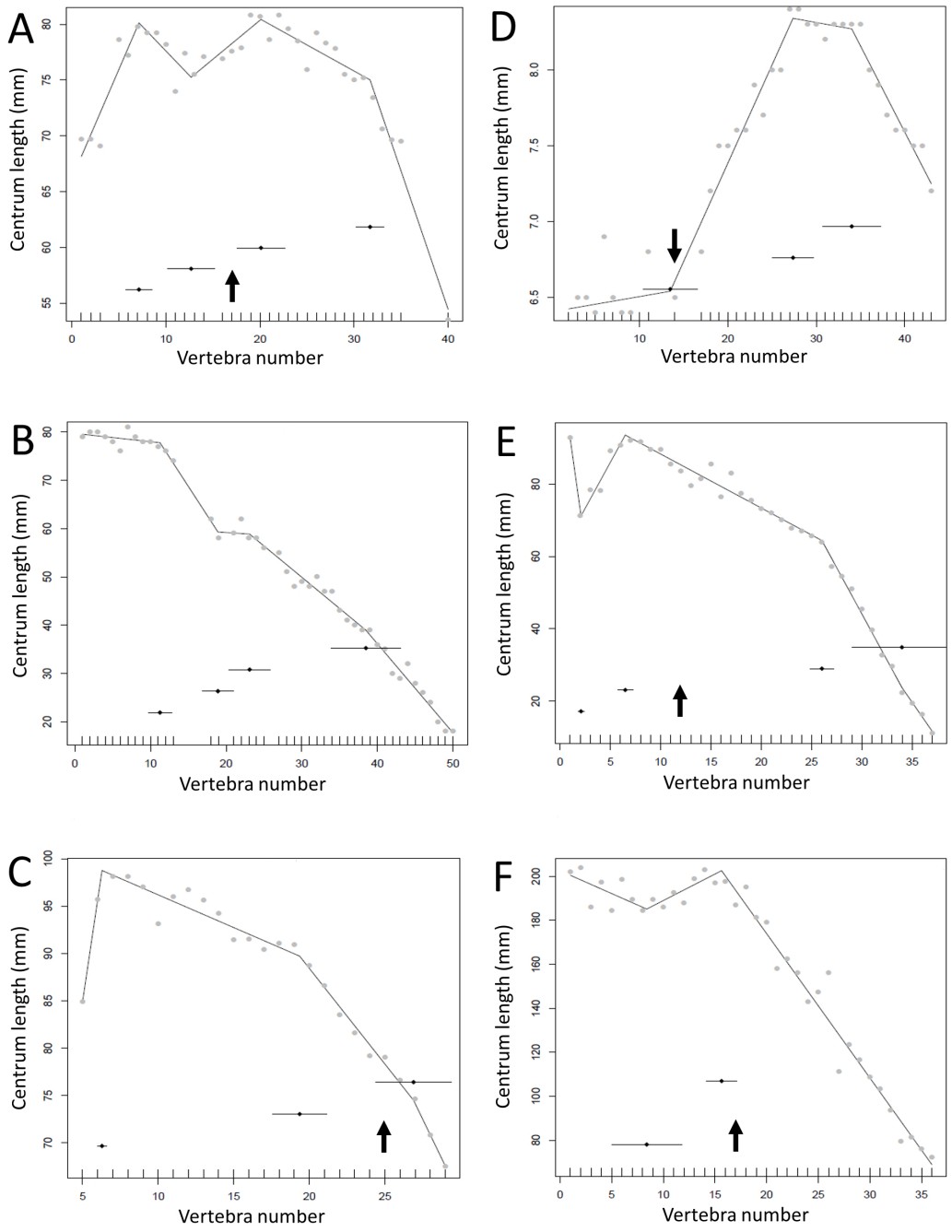

**Figure 7** Regressions for centrum lengths within the tails for non-maniraptoran members of the Theropoda, (A), *Dilophosaurus*, (B) *Ceratosaurus*, (C) *Majungasaurus*, (D) *Juravenator*, (E) *Gorgosaurus*, (F) *Tyrannosaurus*. Note that patterns of the two tyrannosaurs, *Gorgosaurus* and *Tyrannosaurus*, are most similar to each other and that the pattern of *Juravenator* is the most dissimilar from all other theropods. The predicted break points and their error bars are indicated at the bottom of each graph where these have been calculated, and the transition point (where known) is indicated by an arrow.

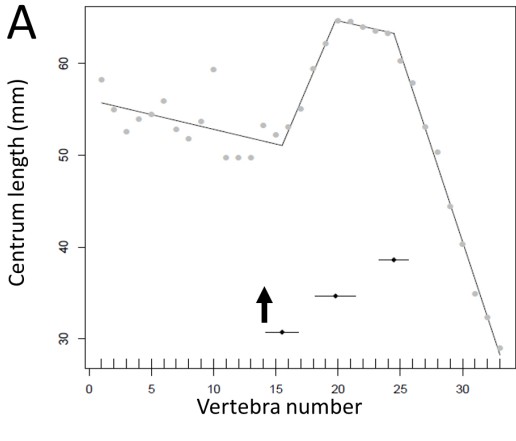

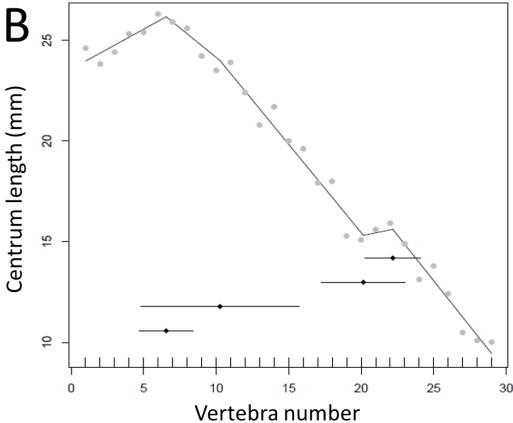

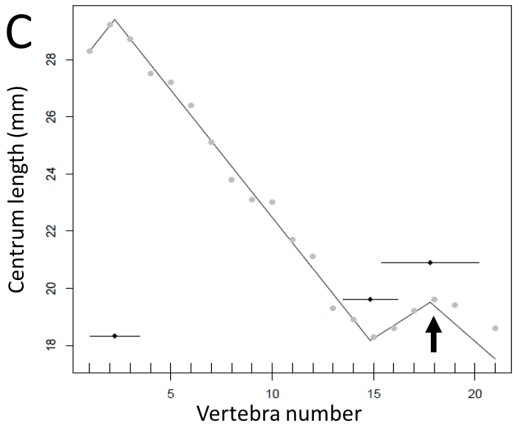

**Figure 8  Regressions for centrum lengths within the tails for selected members of non-paravian Mani-raptora (A)** *Ornithomimus* **(B)** *Ingenia*, **(C)** *Nomingia*. The two oviraptorosaurs (B, C) show strongly tapering tails. The predicted break points and their error bars are indicated at the bottom of each graph where these have been calculated, and the transition point (where known) is indicated by an arrow.

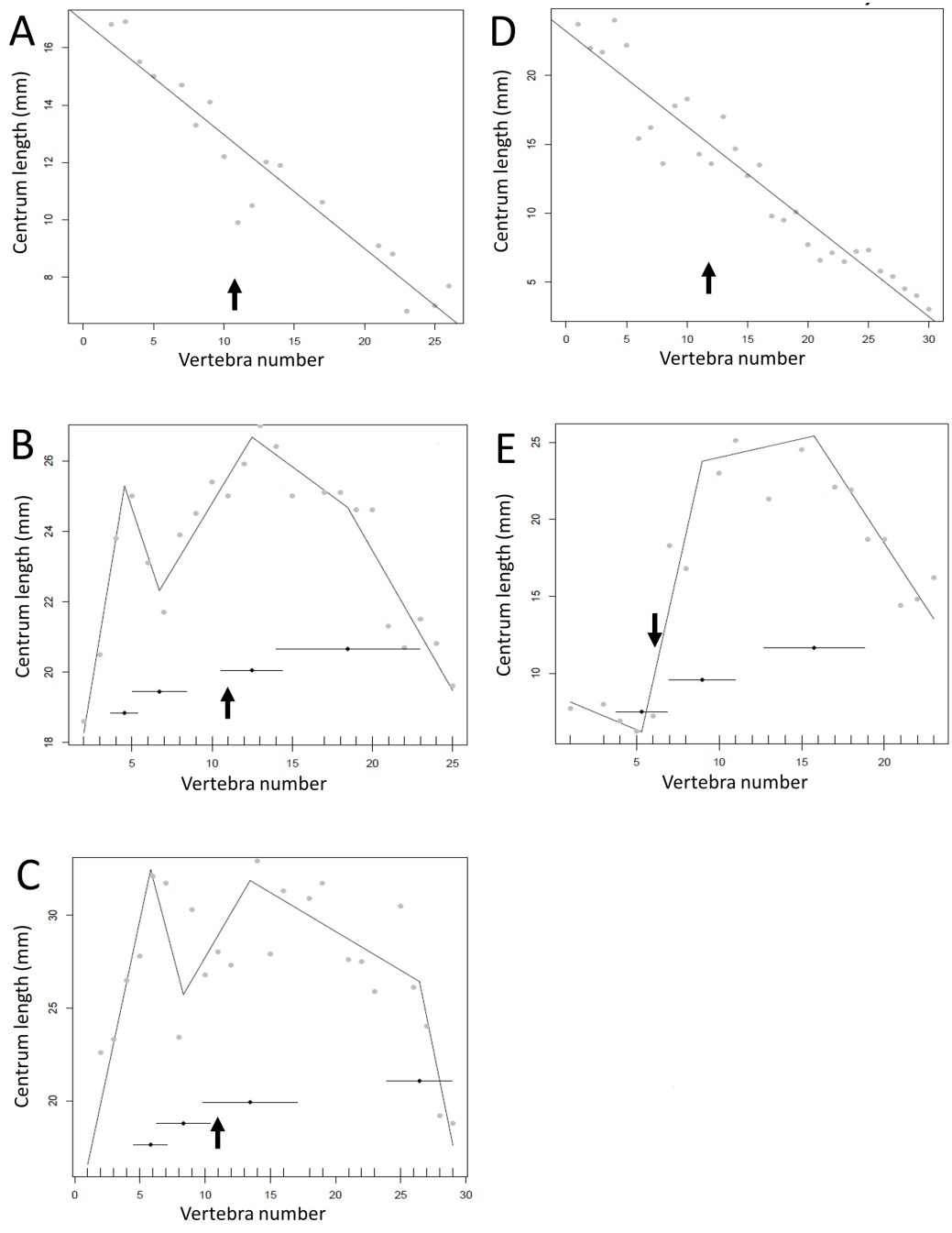

**Figure 9** Regressions for centrum lengths within the tails for members of the Dromaeosauridae, (A) *Velociraptor* IGM 100/25, (B) *Velociraptor* AMNH 100-986, (C) *Velociraptor* MPC 100/985, (D) *Deinonychus*, (E) *Microraptor*. The predicted break points and their error bars are indicated at the bottom of each graph where these have been calculated, and the transition point (where known) is indicated by an arrow.

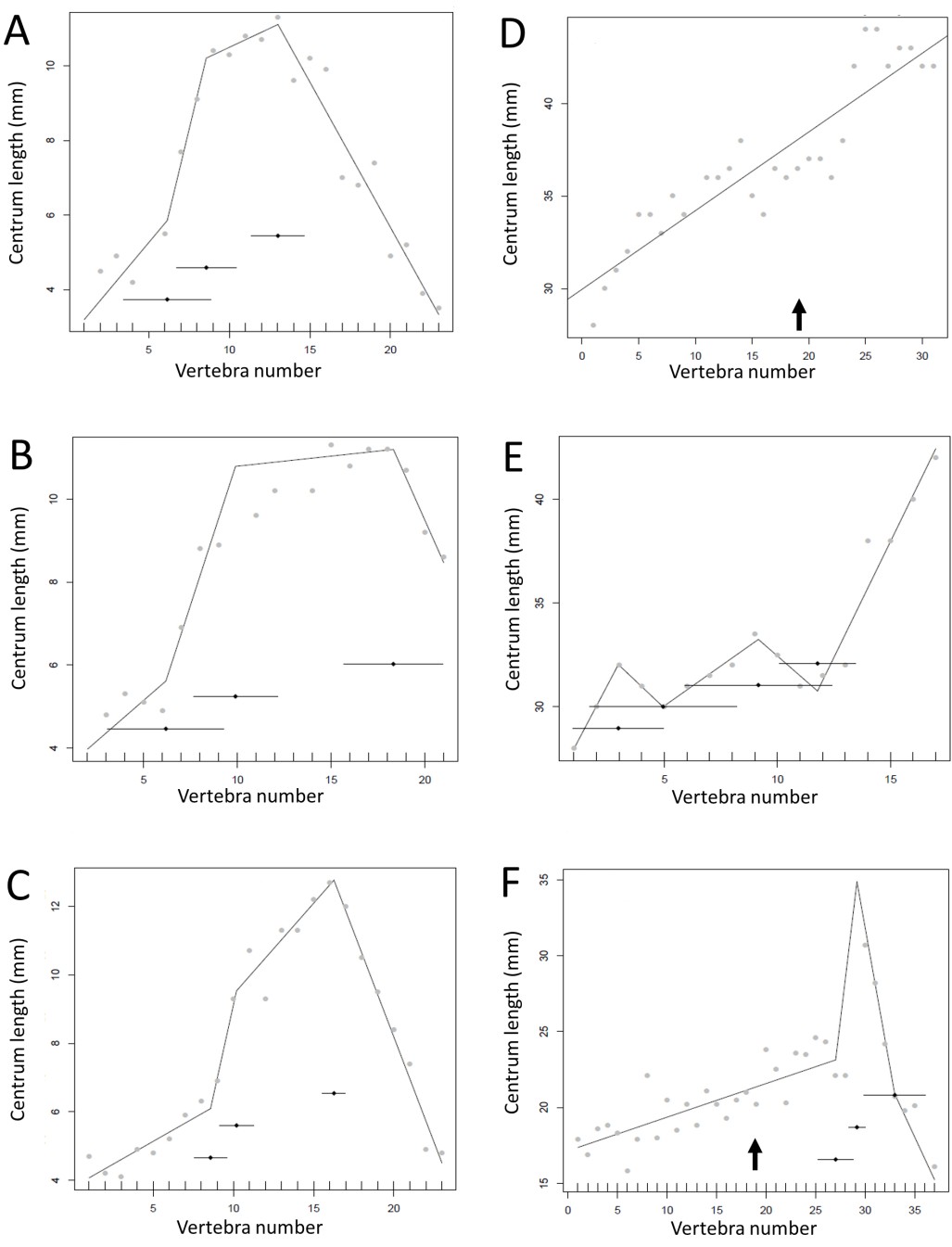

**Figure 10   Regressions for centrum lengths within the tails for specimens of *Archaeopteryx* (A) BSPG1999I50, (B) WDC CGS100 (C) 11th Specimen and *Coelophysis* (D) AMNH 7223, (E) AMNH 7224, (F) AMNH7229.** *Archaeopteryx* is a rare genus for which numerous specimens with complete tails are known. Although the general pattern observed across the specimens is similar, the pattern of each specimen is distinct. Despite all three *Coelophysis* specimens coming from the same locality, their tails differ markedly from each other. The predicted break points and their error bars are indicated at the bottom of each graph where these have been calculated, and the transition point (where known) is indicated by an arrow.

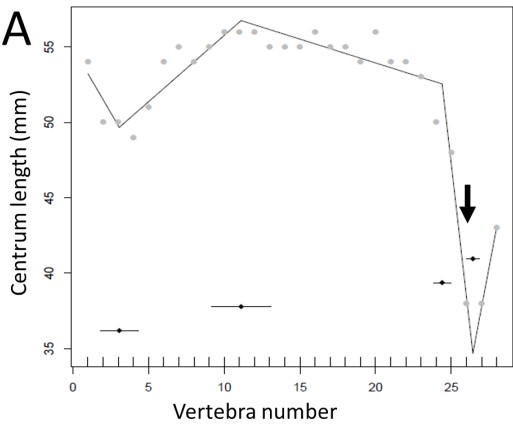

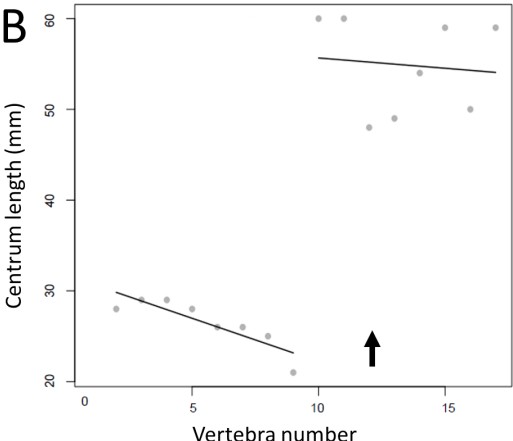

**Figure 11 Regressions for centrum lengths within the tails for members of the Thyreophora, (A) *Kentrosaurus*, (B) *Dyoplosaurus*.** More than any other taxa in the study, the club bearing tail *Dyoplosaurus* would be expected to show a specialized pattern, and it does, although surprisingly the analysis did not recover break points in the series. The predicted break points and their error bars are indicated at the bottom of each graph where these have been calculated, and the transition point (where known) is indicated by an arrow.

flight control from a stiff tail that was only flexible at the very base, and such a condition is reflected here by a proximal section of short centra and then a sudden jump to considerably longer centra.

*Juravenator* is notable as it differs greatly from most other taxa in its pattern of centra lengths (Fig. 7). The specimen is a young juvenile, and it is possible that the tail changes during ontogeny, although *Chiappe & Göhlich (2010)* noted that the caudal length pattern of *Juravenator* (a stable series, then a series of short centra, then a series of long centra, and then a series of shorter centra again) may be consistent across at least some compsognathids. This suggests that *Juravenator* is perhaps not just an outlier, but is representative of a pattern that is normal for its clade. Still, it remains unusual compared to most other dinosaurs.

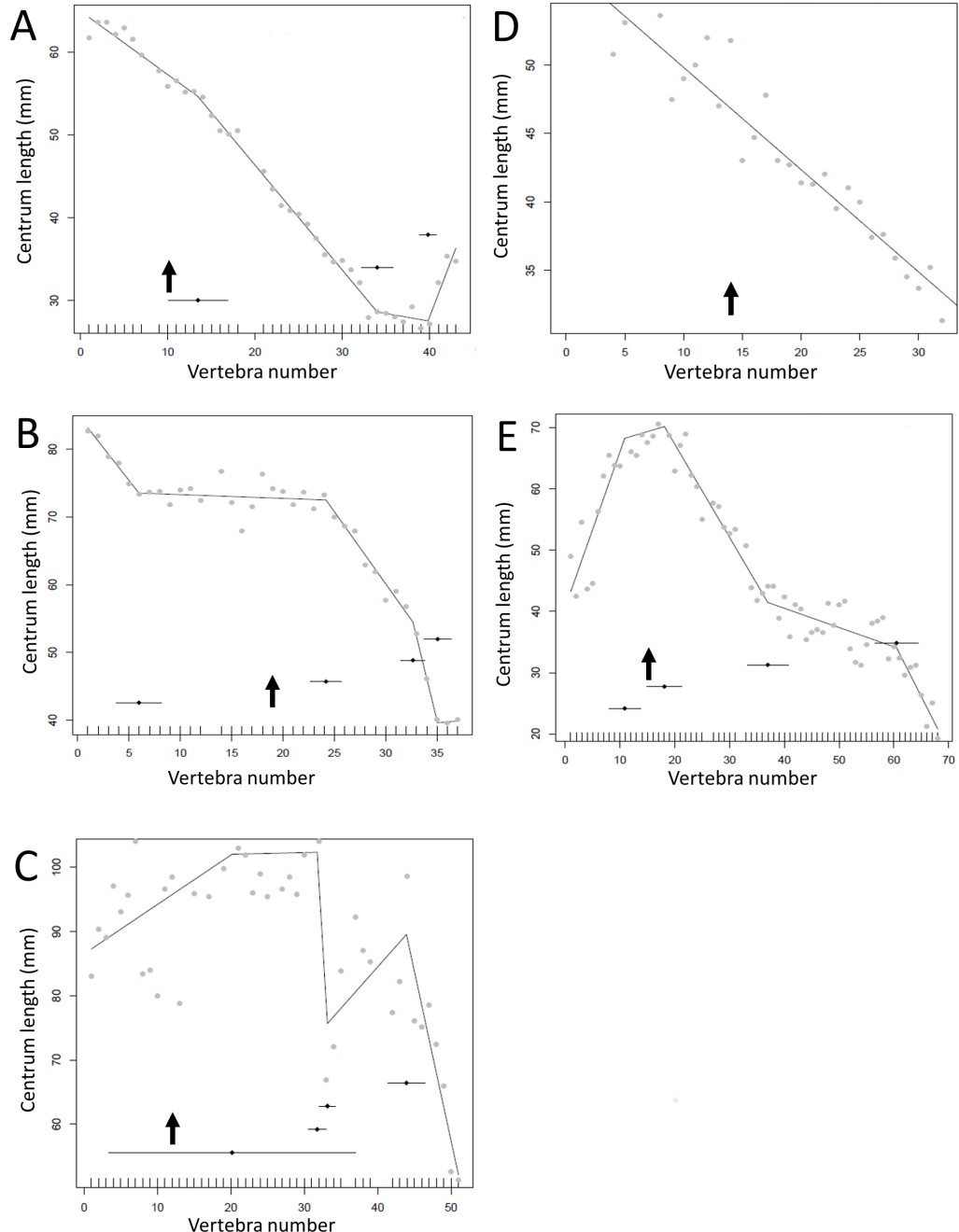

**Figure 12** **Regressions for centrum lengths within the tails for members selected non-hadrosauroid iguanodontians, (A)** *Tenontosaurus,* **(B)** *Ouranosaurus,* **(C)** *Iguanodon* **and hadrosaurs, (D)** *Tethyshadros,* **(E)** *Lambeosaurus.* Despite their overall similarity in form and ecology there are considerable differences between the patterns seen between these taxa. The predicted break points and their error bars are indicated at the bottom of each graph where these have been calculated, and the transition point (where known) is indicated by an arrow.

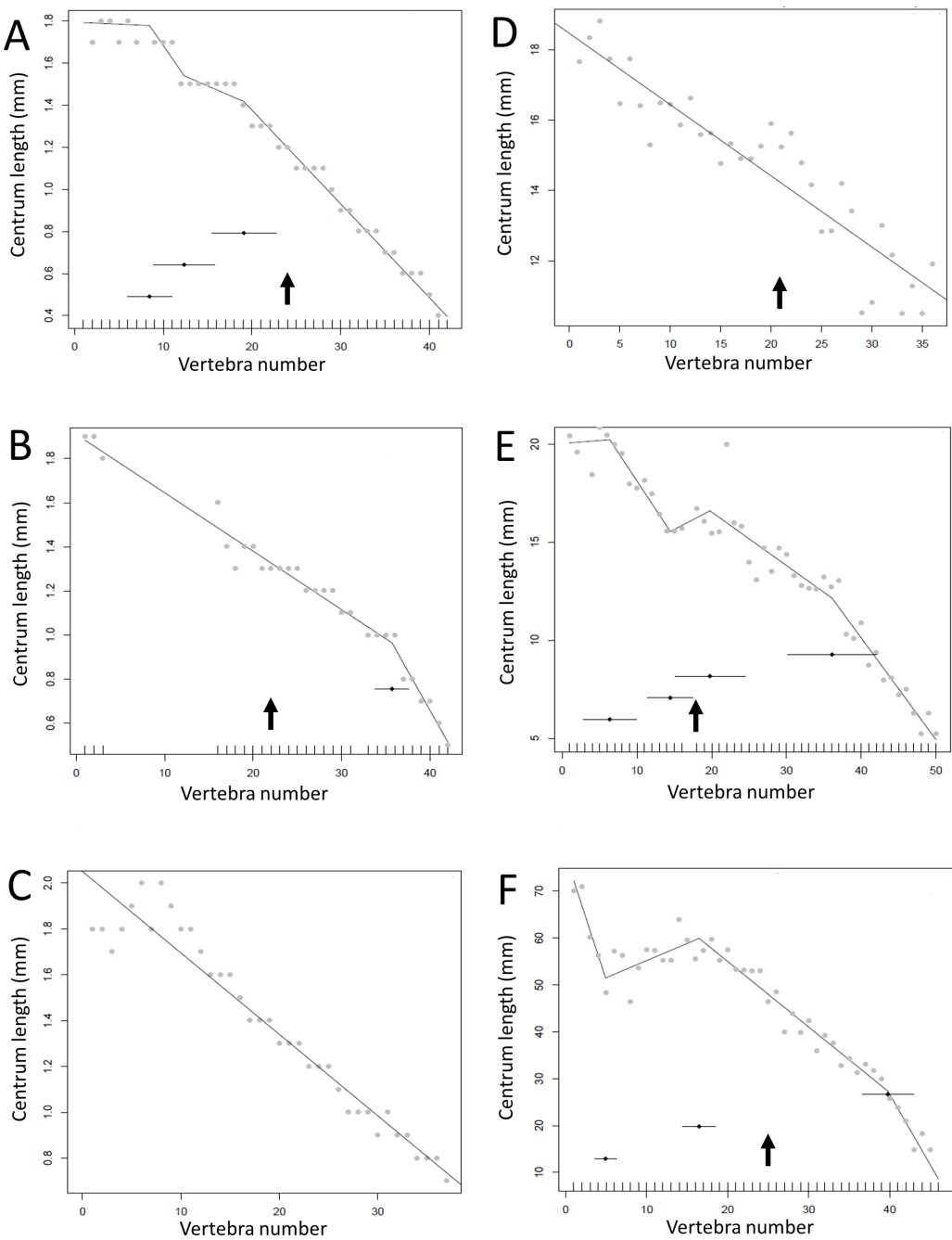

**Figure 13** Regressions for centrum lengths within the tails for members of the Ceratopsia. (A) *Psitta-cosaurus* AMNH 6253, (B) *Psittacosaurus* AMNH 6254, (C) *Psittacosaurus* GiSPS 100/606, (D) *Lepto-ceratops* CMN 8887, (E) *Leptoceratops* CMN 8888, (F) *Centrosaurus.* Relative to that of most other di-nosaur groups considered, the ceratopsians pattern is closer to simple progressive tapering. The predicted break points and their error bars are indicated at the bottom of each graph where these have been calcu-lated, and the transition point (where known) is indicated by an arrow.

## Error

Given the limited availability of data and the problems associated with sourcing information from the literature, mounted specimens, or those with poor preservation, there is likely to be some error in the data. Distortion, if systematic within a specimen, would still preserve the pattern overall and if random (within or between specimens) should have limited overall effect. This is similar to any issues of measurements taken from the literature where different authors may have used slightly different metrics to take the length of the caudal centra, but we would expect consistency within specimens and thus preservation of patterns. As noted above, our own assessment of the possible differences between the measurements from a specimen and from a photograph produced similar results, especially for the prediction of the transition point which was identical in both and correctly identified as caudal vertebra 12 (Fig. 3).

The vertebrae of smaller specimens may cause problems, as these will be harder to measure with equivalent accuracy. For example, the long sequence of identical values seen in *Juravenator* could be because the animal is still a juvenile or may genuinely reflect the unusual caudal anatomy of this genus or of compsognathids as a whole. However, it could simply be because the vertebrae are so small that variation between them (even when attempting to measure to the nearest 0.1 mm) was difficult to detect. Similarly, the alternating sequence of long-short-long seen in various specimens of *Apatosaurus* as described above, may simply be size related (it is easier to measure long vertebrae and find clear differences between them). However, these errors in measurements are collectively likely to be unbiased overall and so should not affect the results or general patterns reported here.

There is some consistency within clades, with for example two specimens of *Leptoceratops* (despite differing caudal counts—Fig. 13), three specimens of *Archaeopteryx* (Fig. 10), and two tyrannosaurids (Fig. 7) all showing similar patterns to one another within their respective clades, suggesting consistency in the data and the analysis. Two specimens of *Velociraptor* are very similar, although a third is rather different, but similar to its near relative *Deinonychus* (Fig. 9).

## Implications

Assuming equivalency of vertebral articulation, for a given unit of tail length, more joints will increase flexibility and fewer joints will reduce flexiblity. Thus, shorter centra imply greater zonal flexibility and longer ones, greater zonal stiffness. As such, we can use the varying lengths of centra within different parts of dinosaur tails to infer differing levels of zonal flexion.

The repeated pattern of a series of short centra, then a series of long ones, and then a series of tapering caudals suggests that many dinosaurs had a flexible tail base, then a stiffened section and finally a more flexible section. Our hypothesis that the transition point (i.e., the termination of the attachment of the caudofemoralis musculature) of dinosaurs is linked to a major change in tail function (as implied by centrum length) is supported by our analyses. We recorded the last centrum to preserve a lateral process from specimens directly or where they were recorded in the literature. Comparing these to the data on

serial variation shows that the transition point often coincides closely with a shift in centra length, (Fig. 4) as determined by the changes in regression lines (e.g., *Lambeosaurus*, Fig. 12, *Juravenator, Gorgosaurus, Tyrannosaurus*, all Fig. 7, *Ornithomimus*, Fig. 8).

Even in taxa where this pattern does not hold, there is evidence that the transition has an influence on caudal length, as the point recorded may be associated with a slight 'hump' in the data (i.e., a short increase and then decrease in vertebral length over just four or five vertebrae), as seen in e.g., *Plateosaurus* (Fig. 5) and *Nomingia* (Fig. 8). As the transition point is perhaps better thought of as a 'zone' (*Persons & Currie, 2011b*), a little leeway must be allowed and we would not expect a perfect correlation between the change in centrum length and the last vertebra possessing a lateral process. Even so, it is clear that the calculated break points in the series often fall exactly on, or within one or two vertebrae of, the last centrum with a lateral process. Based on this, it may be possible to deduce the transition point in some specimens, even if the lateral processes and chevrons are missing or damaged, by interpreting the pattern of centrum lengths. The series of long vertebra that correspond to most of the length of the anterior tail up to the transition point would make the tail relatively stiff, since, in the absence of other anatomical changes, longer vertebra lead to a reduction in intervertebral flexure points per unit of length. This would improve the efficiency of the caudofemoralis muscles as this stiffness would reduce energy loss though lateral movements between vertebrae and ensure that most of the effort of contracting the muscles led to pull on the femur and not lateral tail flexing.

Posterior to the transition point, the requirement for great lateral stiffness is presumably relaxed and thus leads to the simple pattern of general reduction in size of successive centra moving posteriorly. Many taxa independently evolved post-transition-point adaptations for increasing caudal stiffening, such as the relatively elongated centra and extensive zygopophyses of tetanurans, the caudal rods of dromaeosaurs, the pygostyles of various maniraptorans and the handle of ankylosaur tail clubs. Simple selection for reduction in mass posteriorly (with obvious exceptions such as taxa bearing tail clubs) would lead to the pattern seen here and may be the typical primitive condition for reptilian tails. That is seen here in the data for *Varanus* and *Crocodylus* (Fig. S4), and this pattern was also recorded from some other extant and extinct reptiles, and even synapsids by *Nuñez Demarco et al. (2018)*, suggesting that it is a common feature of early amniotes (though clearly this is a very limited dataset).

We suggest that the series of short centra at the base of the tail would allow the entire tail to flex as a unit, as any anterior motion of the tail would also affect any more posterior portion of the tail. Thus a flexible section at the very base of the tail allows the entire tail to be moved without compromising the stiffness of the successive section. This is therefore likely a trade-off between flexion and stiffness. The most proximal section of the tail may also have been less strongly influenced by contractions of the caudofemoral musculature, because the most proximal caudal vertebrae typically lack chevrons, which contribute to the origin of the caudofemoralis. This would afford far less muscle attachment than in the stiffened, middle section of the tail. Additionally, the bending moment of the caudofemoralis would be proportional to the small sine of the angle between the origins of

the caudofemoralis and its lines of action, making the most anterior vertebrae less strongly influenced by this action.

## Evolution of dinosaurian caudal centra series

The short-long-decreasing pattern was likely acquired independently in multiple lineages of dinosaurs. We hypothesise this based on the number of Triassic and / or basal forms (*Plateosaurus*, Fig. 5; *Coelophysis*, Fig. 10), that lack the pattern and present the apparently primitive archosaur (or even diapsid) condition of a simple progressive decrease along the length of the tail. The repeated evolution of various osteological structures able to passively stiffen the tail (e.g., elongate zygopophyses and chevrons in some theropods and ankylosaurs, hyposphene-hypantra articulations in some sauropods and alvarezsaurs, ossified tendons in ornithischians) suggests that tail rigidity was favoured repeatedly in various groups within the Theropoda, Sauropoda and Ornithischia. Thus, the apparent distribution of the short-long-decreasing pattern seen here in later theropods, sauropods and ornithischians may have also arisen independently from similar selective pressures favouring a small zone of high flexibility immediately posterior to the hips and an extended zone of stiffness that helped improve locomotor efficiency.

Despite the wide variations in patterns of elongation and constriction in centrum lengths, it is clear that in at least some cases where there is a reasonable number of caudals preserved and their positions known, it may be possible to reconstruct the size of missing vertebrae with some confidence. Repeated patterns within and between taxa, and long strings of caudals with a consistent pattern of elongation or reduction means that the sizes of missing vertebrae may be estimated. Potentially, even the total length of a tail may be estimated if much of the series is preserved. However, in general this is likely to be difficult –the variation seen here in the patterns of increases and decreases, and the differing numbers of vertebrae in those various sets of increases and decreases are highly variable and difficult to predict. In particular, guessing at what number of vertebrae a tail will end is difficult. While clearly any regression of caudal size that was decreasing successively would eventually suggest a centrum of near zero or negative length, at what point before this the tail would actually terminate cannot be estimated. *Gilmore (1936)* notes with relation to the 'whiplash' segment of the distal tail of *Apatosaurus* that "the uniformity in size of these terminal rod-like caudals is such that any loss would be difficult to detect". Similarly, pygostyles of some theropods (which can reduce very rapidly in size along their length) could also be misleading, as, if only the proximal part were preserved, it would erroneously suggest a much longer tail.

Recognizing variation in caudal counts across taxa has implications for other areas of research. For example, knowing how many vertebrae and of what size ranges a given taxon has may be important to taphonomic analyses that consider sorting or loss of elements by size. In an analysis of a bone bed dominated by the hadrosaur *Amurosaurus*, *Lauters et al. (2008)* looked at the different numbers of element types preserved. They suggested that the vertebrae of *Amurosaurus* were underrepresented in the bonebeds based on the estimated number of vertebrae in the axial column. However, with very little articulation known for remains of *Amurosaurus*, it cannot be easily estimated how long the tail was or

how many caudals it possessed, with tails for hadrosaurs known to have as few as around 50 caudals (*Horner, Weishampel & Forster, 2004*) to over 75 (*Lull & Wright, 1942*). (There is of course also likely variation within the number of the cervical, dorsal and sacral series, though based on *Hone (2012)*, this is likely to be much less of an issue than the caudal series). The results of *Lauters et al. (2008)* for *Amurosaurus*, were robust and in this case such an issue is not likely to have had a major effect on their results, but the uncertainty surrounding the number of vertebrae in the axial column means that care should be taken when performing such an analysis. We suggest that, unless the true axial count is known with confidence, either caudals should not be counted, or upper and lower estimates of the number of vertebrae in the column should be employed, or such considerations should be limited to the lateral process bearing caudals.

## CONCLUSIONS

Total tail length remains difficult to estimate for incomplete tails in the Dinosauria, and there is strong variation both between and within Family rank equivalent clades. However, there is some consistency in patterns of individual centra length within tails. Notably, the proximal part of the tail often consists of relatively short vertebrae, followed by a series of longer vertebrae and then a shift to decreasing centrum lengths beyond the transition point. This pattern suggests an underlying constraint in dinosaurian tail function, where (1) the very base of the tail was flexible and allowed the large remaining posterior portion of the tail to be swung as a collective whole, (2) shortly past the tail base there was a zone of relative stiffness that supported the caudofemoralis musculature, and (3) after the termination of the caudofemoralis and for a highly variable distance, the remaining vertebrae tapered to a reduced size.

While length of centrum is the most relevant measure of an individual vertebra's contribution to total tail length, numerous other traits have the potential to influence basic aspects of caudal function, including lateral flexibility and the functioning of the caudofemoralis. Future studies of variation patterns in other traits (such as centrum width and zygapophyseal articulation) will refine these general observations. Although developed in order to investigate dinosaur tails, the methods used here to separate out groups of similar units as part of a series may be widely applicable to assessments of any extinct and extant taxa exhibiting repeating anatomical units (e.g., the dimensions of any vertebral series, scalation, or body segments in annelid or arthropod invertebrates).

**Institutional abbreviations**

| | |
|---|---|
| **AMNH** | American Museum of Natural History, New York |
| **BMNS** | Belgium Museum of Natural Sciences, Brussels |
| **CAGS** | Chinese Academy of Geological Sciences, Beijing |
| **CM** | Carnegie Museum of Natural History, Pittsburgh |
| **CMN** | Canadian Museum of Nature, Aylmer |
| **CYGYB/CYNG** | Chaoyang Paleontological Museum, Chaoyang, Liaoning |
| **FMNH** | Field Museum of Natural History, Chicago |
| **GIN/Gi-SPS** | Institute of Geology, Mongolian Academy of Sciences, Ulan Baator |

| | |
|---|---|
| GMZ | Grant Museum of Zoology, London |
| IGM | Mongolian Academy of Sciences, Ulan Baator |
| IVPP | Institute of Vertebrate Paleontology and Paleoanthropology, Beijing |
| JME | Jura Museum, Eichstätt |
| JMP | Henan Geological Museum, Henan Province |
| LPM | Liaoning Provincial Museum of Paleontology, Liaoning |
| MB.R. | fossil reptiles collection of MfN (Museum für Naturkunde) Berlin |
| MNA | Museum of Northern Arizona, Flagstaff |
| MPC | Mongolian Paleontological Centre, Mongolian Academy of Sciences, Ulan Baator |
| NHMUK | British Museum of Natural History, London |
| OMNH | Oklahoma Museum of Natural History, Norman |
| PIN | Paleontological Institute, Russian Academy of Sciences, Moscow |
| PMOL | Paleontological Museum of Liaoning, Shenyang Normal University, Shenyang |
| QM | Qijiang Dinosaur National Geological Park Museum, Liaoning |
| ROM | Royal Ontario Museum, Toronto |
| RTMP | Royal Tyrrell Museum of Palaeontology, Drumheller |
| SC | Italian State Collections |
| SMA | Sauriermuseum Aathal, Aathal |
| UCMP | University of California Museum of Paleontology, Berkeley |
| USNM | Smithsonian Museum of Natural History, Washington, DC |
| YPM | Yale Peabody Museum, New Haven |
| ZDM | Zigong Dinosaur Museum, Zigong |

## ACKNOWLEDGEMENTS

We wish to thank the numerous colleagues who provided papers, discussions, and details of specimens with complete tails that are buried in the literature and research collections: Jordan Mallon, Matthew Herne, Susie Maidment, Corwin Sullivan, Fabio Dalla Vecchia, Andrea Cau, Victoria Arbour, Mickey Mortimer, Phil Currie, Matt Wedel, Pascal Godefroit and Chris Rogers. For access to specimens we thank Sandra Chapman and Fang Zheng. We thank Mike Habib for discussion of tail evolution and stiffening and Rob Knell for assistance with the statistics. We also thank Victoria Arbour, Heinrich Mallison, Eric Snively, Susie Maidment and David Polly for comments on an earlier version of this manuscript and Mark Young for handling it as editor.

### Funding

The authors received no funding for this work.

### Competing Interests

The authors declare there are no competing interests.

## Author Contributions

- David W.E. Hone and W. Scott Persons conceived and designed the experiments, performed the experiments, prepared figures and/or tables, authored or reviewed drafts of the paper, and approved the final draft.
- Steven C. Le Comber conceived and designed the experiments, analyzed the data, prepared figures and/or tables, authored or reviewed drafts of the paper, and approved the final draft.

## Data Availability

Raw measurements of specimens are available in the Supplemental Files.

Specimens are archived at the following museums (this information is also available in the Supplemental Files):

- American Museum of Natural History, New York: AMNH 223, 5351, 6253, 6254, 7229, 7223, 7224
- Bavarian State Collection for Palaeontology, Munich: BSPG 1999I50
- Chinese Academy of Geological Sciences, Beijing
CMN: CAGS IG040801
- Canadian Museum of Nature, Ottawa: 8547, 8887, 8888, 8889
- Chaoyang National Geopark, Chaoyang: CYNG 024
- Carnegie Musuem of Nature, Pittsburgh: CM 3378, 3018, 11338
- Field Museum of Natural History, Chicago, FMNH: PR2100, PR2081
- Geological Institute Section of Palaeontology and Stratigraphy, the Academy of Sciences of the Mongolian People's Republic, Ulan Bator
GMZ unnumbered Grant Museum of Zoology and Comparative Anatomy, London: Gi SPS 100/606
- Humboldt Museum of Nature, Berlin: HMN (also MBR) unnumbered
- Mongolian Institute of Geology, Ulaan Bataar: IGM 100/1002, 100/1127
- Institute of Vertebrate Paleontology and Paleonathropology, Beijing: IVPP V12430, V16055, V12653, V12529, V120888, V14341, V11115, V13352, V15471
- Jura Museum, Eichstätt: JME Sch200
- Jinzhou Museum of Paleontology, Jinzhou: JMP V-05-8-01
- Liaoning Provincial Museum of Paleontology, Liaoning: LPM 0226
- Museum of Comparative Zoology, Cambridge: MCZ 4188
- Museum of Northern Arizona, Flagstaff: MNA PI175
- Mongolian Paleontological Center, Mongolian Academy of Sciences, Ulaanbaatar: MPC 100/1305, 100/30, 100/119, 100/985
- Natural History Museum, London: NHM R1111
- Sam Noble Oklahoma Museum of Natural History, Norman: OMNH 63525, 34191
- Palaeontological Institute, Russian Academy of Sciences, Moscow: PIN 614
- Palaeontological Institute Shanyang Normal University, Shenyang
QM V1002Qijiang Dinosaur National Geological Park Museum, Qijiang: PMOL 0006
- Royal Ontario Museum, Toronto: ROM 804, 845, 1218

- State Collection of Italy, Rome: SC, 57021
- Aathal Dinosaur Museum, Zurich: SMA 0010, 0092
- Royal Tyrrell Museum of Paleontology, Drumheller: TMP 90.26.01, 1995.11.001, 1998.58.01
- University of California Museum of Paleontology, Berkeley: UCMP 37302
- Smithsonian Museum of Natural History, Washington DC: USNM 4735
- Wyoming Dinosaur Centre, Thermopolis: WDC CSG100
- Yale Peabody Museum, New Haven: YPM 1884, 5023
- Zigong Dinosaur Museum: ZDM 5050
- Institute of Paleobiology, Polish Academy of Sciences, Warsaw: ZPAL MgD-I/48

## Supplemental Information

Supplemental information for this article can be found online at http://dx.doi.org/10.7717/peerj.10721#supplemental-information.

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
