# Peer review of "New data on tail lengths and variation along the caudal series in the non-avialan dinosaurs"

_PeerJ, doi:10.7717/peerj.10721_

## Round 0.1 · original submission · Major Revisions

Dear authors,

As the three reviewers all gave different decisions, I have made the editorial decision of 'major revisions'. This is due to the points raised by reviewer two, and the number of comments between all three reviewers.

I agree with reviewers two and three that your hypothetico-deductive approach could improved. Please also pay special attention to the figure comments by reviewer two. I think the comments from all three are very helpful and constructive, and I do not foresee any problems addressing them.

I look forward to receiving your revised manuscript.

·

Basic reporting

In some parts the text is a bit colloquially phrased; I have marked a few instances. Please go through the entire MS, not just the marked instances.

Experimental design

no comment

Validity of the findings

no comment

Additional comments

This MS was a joy to read, as the structure and language are clear, the relevant literature has been taken into account, the figures are sufficient and clear (exception: in on the species names overlap; if possible this problem should be correct - not essential, though, as the relevance of the individual labelled values is low and the relevant aspect, the curve, is clearly visible), the data is well presented, and the hypotheses are clearly defined and tested.

Aside from some minor text editing, this is in my view ready for publishing!

·

Basic reporting

See general comments.

Experimental design

See general comments.

Validity of the findings

See general comments.

Additional comments

Thanks for the opportunity to review this manuscript. I think the main observation of the paper, that dinosaur caudal vertebral length does not just uniformly decrease along the length of the tail, and the resulting functional implications, is interesting. However, I have several issues with the structure of the paper and the approach taken that I think need addressing before the paper could be accepted for publication.

1. Introduction and rationale for the study: I do not find the hypothesis to be clearly laid out and the results and discussion do not link really clearly back to the stated hypotheses and predictions. What are you really testing, because it feels like more than one thing, which is fine but should be specified – I feel like questions you are trying to answer are:
• Do caudals smoothly decrease in length, yes or no?
• If no, is there a consistent pattern across dinosaurs or at least some dinosaur groups?
• What do the patterns mean?

Or, are you asking:
• Can you reconstruct body length from tail length alone?
• Can you reconstruct fragmentary tail lengths from more complete specimens?

I think the introduction could be strengthened by a more in depth discussion of caudal anatomy in other tetrapod groups and its functional implications, and how that relates to dinosaur caudal anatomy. Have caudal length patterns been investigated in other groups? What approaches were taken? What were the results?

2. Methods: I’m not sure the methods used are appropriately addressing the hypothesis. I feel like what you want to do is use AIC to determine whether a single segment model or a multiple segment model better fits the data on dinosaur tail lengths, which would test your hypothesis that caudals do not just smoothly decrease in length, but I don’t think that’s exactly what you’ve done.

I’m also concerned that the analysis is not picking up all of the morphology, because I don’t understand the graph presented for Dyoplosaurus. Ankylosaurids have a series of short, ‘typical’ dinosaur caudal centra anteriorly and then those in the tail club are abruptly much longer at about the midpoint of the tail. Surely there should be a break point indicated at the beginning of the tail club, but it doesn’t look like that’s the case in the graph? If so, can you comment on why that might be, and what implications that has for the other graphs and your interpretations?

I think it would also be very beneficial to look at the phylogenetic signal of caudal centrum length patterns or femur to tail length ratios, since one of the points made in the paper is that you can’t use tail length to predict other aspects of anatomy. Would also be helpful for interpreting how easy it is to reconstruct lengths of partially complete tails.

3. General paper structure: Very few results are actually presented in the results section, and a lot are presented in the discussion, which needs rectifying. The results section has a lot of very vague sentences with broad generalizations and not a lot of reporting of statistical results or qualitative patterns. The results section should detail caudal anatomy and the results of your statistical analyses; the discussion should interpret those results in terms of your hypothesis, the functional implications, etc. I also noticed quite a few typos throughout the manuscript, and think it could benefit from a very close read-through for both spelling and sentence readability.

4. Figures: I do not think the first figure adds much to the paper, especially because no anatomical features of note are labelled on it. At minimum, labels should be added to this, but a really useful figure would probably include line drawings of the major clades described here.

The figure detailing the repeated measurements for the hadrosaur specimen vs photograph measurements should be Figure 2, since this is a proof of concept for your use of photograph measurements.

The remainder of the results figures could probably be condensed into fewer individual figures, and it would be helpful for them to be organized in some kind of phylogenetic structure, e.g. all theropods together or at least in a few figures, all sauropods together, all ornithischians together or near each other. The legend describing the break points and transition point should be in the caption text for each relevant figure, not just one. Full species names and specimens should also be included either in the graphic of the figure or in the caption, please don’t make people spend a bunch of energy figuring out which specimen you’re talking about by going through your appendix.

I’m having a hard time parsing Figure 3, is there another way this information could be presented? Or could additional information be included in the caption that makes it easier to understand?

Why does figure 6 appear where it does? Should this be before or after the bulk of the other data figures? Figure 6 is also very difficult to interpret because all of the taxon names are overlapping the data, please clean this up so that it’s more legible.


I have also attached an annotated PDF with some additional notes and comments throughout the text. I hope these comments help to strengthen the revised version of the manuscript.

-Victoria Arbour, 9 May 2020

·

Basic reporting

The manuscript upholds standards for basic reporting, and will be enhanced with just a few improvements to mechanics and content.

Avoid and rework "due to" clauses for slightly more active and direct communication.
Replace "nature of" with specific concepts.
Re-phrase at least two sentences that separate subject and verb with a minor avalanche of (necessary) prepositional phrases.

Consider Gauthier (1986) and Tetanurae when discussing transition points.

For each figure caption, include a take-home message that you want the reader to understand from the visuals.

Experimental design

Other reviewers can better comment on the statistics. For R plots, consider abbreviations rather than long genus names that clutter the chart.

When justifying why centrum length is interesting, briefly acknowledge other dimensions by citing Currie (1978) on centrum transverse radius.

Groups for comparison are well-chosen, but the hypotheses are rather dry. To better frame the question, reiterate "why" after you state the hypotheses.

Impressive venture comparing physical and photograph-based measurements.

Validity of the findings

The most interesting results relate to variation within and between dinosaurian clades and locomotor categories. The exploration was intriguing, and sets up questions of functional morphology.

Clarify earlier which parts of the proximal region of tail are stiff and therefore useful for bracing against m. caudofemoralis longus action. Gauthier (1986) considered the distal region of the tail in theropods (posterior to the transition point) to act as a stiff, unitary dynamic stabilizer.

It's unclear why long centra just posterior to the base of the tail would limit its flexibility in this region. What if joints between these long centra were unconstrained ball-and-socket configurations? Posit how you might further test this flexibility.

The conclusions about applicability to other segmented animals seem dropped in as a hopeful afterthought. Try to tie these better to your methods.

Additional comments

The attached commented manuscript elaborates on points in this review, and includes specific suggestions. Figure 1 is welcome, but more images of actual dinosaur tails, and/or thumbnails of full skeletons, would illustrate the animals behind the uniform or regional regressions. I suggest at least one figure showing tails in various groups or tail thumbnails in each graph, and a figure with skeletons illustrating locomotor categories.

As mentioned under "basic reporting", elaborate captions with take-home points for each figure. Lots of readers will jump first to the visuals and captions. What do you most want them to notice and learn from a figure? The caption for Figure 1 does this quite well.

---

## Round 0.2 · Major Revisions

Dear authors,

I agree with reviewer two’s reccomendation of ‘major revisions’. Please give careful consideration to their comments.

I look forward to receiving your revised manuscript.

·

Basic reporting

This paper has improved a lot, especially in areas (statistic) outside my area of expertise. Overall, I find the authors' response convincing with regards to the review recommendations they chose not to follow. The grammar has been improved, colloquial phrasing has been removed.

Experimental design

all fine

Validity of the findings

all fine

·

Basic reporting

See general comments.

Experimental design

See general comments.

Validity of the findings

See general comments.

Additional comments

Thanks for providing the revised manuscript and author comments for a second round of review. In my first review I noted issues with the introduction (ease of understanding the hypotheses, background information from other literature), methods, general paper structure, and figures. I’ve structured my current round of responses into the same categories and note where I feel comments have been satisfactorily resolved vs. where I think additional work is needed. I have extensively marked up the attached PDF and have not repeated all of my comments here, so please review the comments on the PDF as well.


1. Introduction and rationale for the study.
In the response to my comments the authors wrote: “With respect to the point about other taxa, we think this is well beyond the scope of this paper since while clearly relevant, this is an attempt to produce and analyses data on dinosaur tails…Studies of reptile tails we have looked at have focussed on flexibility (joints, articulations) and particular as anchors for key muscle groups and the effect on propulsion and there is almost nothing on total length, or the lengths of centra or patterns of centra.”
I actually think this is an incredibly valuable thing to note because it highlights the novel aspects of this manuscript. I don’t see why you can’t have a sentence that says something along those lines – author X has looked at joints/articulations/flexibility but not about length patterns across multiple families. That is helpful!

Overall I think the hypotheses are a bit more clearly laid out but (as I’ll note in detail shortly) I think the introduction could use another close edit for things like paragraph structure, readability, and conciseness (see general paper structure section).


2. Methods:
I am satisfied with the response to my comments about AIC and Dyoplosaurus. However, a follow-up question is whether or not any other taxa in the dataset should have been treated as discontinuous? I think the answer is reasonably ‘no’, but it would be good to explicitly state why *only* Dyoplosaurus was treated this way, and if you can provide statistical support for one method vs the other that would obviously be beneficial.

In my last review I had noted that it would be helpful to look at phylogenetic signal since a large part of the discussion is about reconstructing caudal anatomy or predicting body size based on tails. While I understand that the data is skewed and it would be extra work to give this a try, I think you need to be very cautious about how you frame your results about reconstructing tail lengths/body size in the absence of phylogenetic context. At minimum, I think you should include a note about why you didn’t look at phylogenetic signal for this – noting that the data is restricted and skewed would be sufficient.


3. General paper structure:

I think the issue with results being reported in the discussion section and such is basically resolved, and that this is much improved from the first submission. However, I think there is still a substantial amount of revision needed in order to tighten up the manuscript overall. For example:
• the Results section does not detail the results of the photograph vs. specimen measurement comparison
• there seem to be contradictory statements at various points in the Discussion about whether you can or cannot (or should or should not, I guess) reconstruct missing elements from partial tails
• there are numerous typos and grammatical glitches that don’t seem to have been resolved from the previous version or might have been introduced when writing new text

In my previous review I noted that the manuscript needed a close read-through for spelling and sentence readability, and unfortunately I do not think there was much improvement on that front in this draft. I found overall that I was having a hard time following the flow of arguments in this paper because there are a lot of parenthetical statements, sequences of sentences beginning with ‘thus’, ‘therefore’, ‘in other words’, etc., and qualifying words like ‘likely’, ‘possibly’, etc. I’ve tried to make some suggestions for where sentences can be trimmed down and made more concise without losing the meaning of the sentence, and which hopefully make the point of the sentence more impactful. I cannot edit the entire manuscript in this way and so I really, really encourage the authors to go back through the ms with a metaphorical fine-toothed comb, make sure all the paragraphs flow well both internally and from paragraph to paragraph, and see if sentences can be trimmed down.


4. Figures:
I appreciate the changes to the figure captions throughout and the inclusion of genus names and specimen numbers. However, I still have several major issues with the figures.

Fig 1: The reviewers note that “We think figure 1 is useful (as reviewer 3 notes) in its current form so have not changed it. Adding detailed drawings of the tails of several taxa would be a considerable amount of work and these are fairly easily accessible in other papers with specialist descriptions (which are already cited).” I note that Reviewer 3 specifically noted “Figure 1 is welcome, but more images of actual dinosaur tails, and/or thumbnails of full skeletons, would illustrate the animals behind the uniform or regional regressions. I suggest at least one figure showing tails in various groups or tail thumbnails in each graph, and a figure with skeletons illustrating locomotor categories.” This is almost exactly the same as my request that a range of tails and their varying morphologies be included in the paper and I frankly find it disingenuous to frame this as though Reviewer 3 and I are in opposition on this point when we are clearly not. (I had written “I do not think the first figure adds much to the paper, especially because no anatomical features of note are labelled on it. At minimum, labels should be added to this, but a really useful figure would probably include line drawings of the major clades described here.”)

I am not sympathetic to the argument that adding labels to this figure, or better yet, revising this figure to include images of other dinosaur tails, is too much work. I think in a paper about the morphology of dinosaur caudal vertebrae and their functional significance that it is entirely reasonable to have a figure showing the range of morphologies of clades mentioned in the study, and to label things like the transition point, extent of the caudofemoralis, and centrum/neural arch/haemal arch/transverse processes. Your readers will obviously include seasoned professionals in vertebrate palaeontology, but will also presumably include nonspecialists and students. Don’t make them go wading through the literature just to know what a whiplash diplodocid tail really looks like, or an ankylosaur tail club, or an oviraptorid pygostyle. I am fairly certain the authors have photographs of specimens they have personally observed that could be compiled into a plate of photos that would be incredibly useful to the readers of this paper. It’s probably the sort of figure that would find its way into university lectures on dinosaurs and vertebrate anatomy, and honestly it’s probably the sort of image that would get tweeted out about the paper and help the paper gain a wider readership.

As another point along these lines, I’d like to highlight the fact that your statistical analyses did not recognize ankylosaurid tail clubs and it was my familiarity with their anatomy that caught this, because I have a mental image of what an ankylosaurid tail looks like. Being able to visualize the morphology of the tail was important for vetting the results of the statistical analysis and interpreting the results. Similarly, having photographs or illustrations of tails at hand in the manuscript would be helpful for readers to be able to associate the abstracted data in the plots with the functional interpretations you make in the discussion.

I leave it to the editor to make the call on this one, but obviously I think the communication value of a figure with some pictures of dinosaur tails outweighs concerns about the amount of effort it would take to create that figure. Reviewer 3 clearly seems to have thought along the same lines.

Fig. 6: I was disappointed to see that there were no changes to Fig 6 and that the species names are still overlapping the data points. As I mentioned previously, this figure needs to be cleaned up so that the plotted data is visible. This is probably annoying and difficult to do in R (which I presume the plot was originally made in), but probably pretty easy in Illustrator or the vector program of your choice.

Fig 3 and overall order of figures: I find it weird that figure 3 (about body size) comes after one of the figures showing break points for sauropods, and then we jump back into break point graphs for other dinosaurs. This is because sauropod results are mentioned in the methods section – I think that text needs to be rewritten so that what is currently Figure 2 appears sequentially along with the remaining break point figures. In my last round of comments I said that the figures should be organized in some kind of phylogenetic structure and this is what I was getting at – currently the figures are, in order: anatomy, sauropods, body size, theropods, hadrosaurs, summary of break points, thyreophorans, ceratopsids, sauropodomorphs, theropods, theropods, theropods, ornithopods. This isn’t an order that really makes sense to me, and in reviewing their appearance in the text I don’t see the pattern either. A figure order that makes more sense, to me at least, would be: anatomy, photograph vs specimen measurements (which is not in the current list but is in the SI), body size, thyreophorans, ceratopsians, ornithopods/hadrosaurs, sauropodomorphs, sauropods, theropods, theropods, theropods, theropods, summary of break points. That’s like walking across the Dinosauria tree, and that’s how I’d look for this data if I was flipping through the figures in the paper looking for specific clades. Alternately you could group results by quadrupedal/bipedal taxa but still at least put sauropods and sauropodomorphs closer to each other within the text (they’re currently separated by 7 figures). You could just have a sentence at the beginning of the section where you talk about break points in your Results that says something like “The results of our break point analyses are presented in Figures 3-11” or however many there are.



I continue to think that the results of this paper are interesting and worthwhile to share via publication and I appreciate the work the authors have done to collate this dataset. I offer these critiques in the spirit of improving the manuscript so that the effort that has gone into collecting and analyzing this data is maximally useful to readers and reaches a wide audience.

-V. Arbour, 27 Aug 2020

---

## Round 0.3 · Minor Revisions

Dear authors,

I have accepted the reviewer's decision of 'minor revisions'.

I look forward to receiving your revised manuscript.

·

Basic reporting

See general comments.

Experimental design

See general comments.

Validity of the findings

See general comments.

Additional comments

I have two remaining comments on the manuscript, both of which are minor, and I do not feel that I need to review this manuscript again:

1. Anatomical features of note should be added to figure 1, such as centrum, neural spine, haemal spine, transition point, etc.
2. I am still catching a number of typos and grammatical errors in this revision, which I've highlighted on the attached PDF.

---

## Round 0.4 · accepted · Accept

Dear authors,

Thank you for your revised manuscript. I am happy to say it has been accepted for publication.

You will shortly be contacted by the production staff to take you through the proof stage.

Once again, thank you for choosing PeerJ, and I hope you will use us as your publication venue again in the future.